# Lying, Feeding and Activity Preference of Weaned Piglets for LED-Illuminated vs. Dark Pen Compartments

**DOI:** 10.3390/ani12020202

**Published:** 2022-01-15

**Authors:** Sven Götz, Camille M. C. Raoult, Klaus Reiter, Monika Wensch-Dorendorf, Eberhard von Borell

**Affiliations:** 1Department of Animal Husbandry and Ecology, Institute of Agricultural and Nutritional Sciences, Martin-Luther University Halle-Wittenberg, Theodor-Lieser-Str. 11, 06120 Halle (Saale), Germany; camille.raoult@landw.uni-halle.de; 2Bavarian State Research Center for Agriculture, Institute for Agricultural Engineering and Animal Husbandry, Prof.-Dürrwaechter-Platz 5, 85586 Poing, Germany; klaus.reiter@lfl.bayern.de; 3Department of Biometrics and Agricultural Informatics, Institute of Agricultural and Nutritional Sciences, Martin-Luther University Halle-Wittenberg, Karl-Freiherr-von-Fritsch-Str. 4, 06120 Halle (Saale), Germany; monika.dorendorf@landw.uni-halle.de

**Keywords:** illuminance, weaned piglets, lux, behaviour, animal welfare

## Abstract

**Simple Summary:**

Knowledge about animal welfare-based lighting in pig farms is very limited, as there is little research on this topic. Legal requirements are often not scientifically supported and differ greatly among countries. However, negative effects of uncontrolled lighting on pig health and behaviour are known. In this study, the influence of different illuminance levels on the preference behaviour of pigs was determined. Piglets were given a free choice between two illuminance levels. We found that over time piglets preferred darker pen compartments to lie down and brightly lit ones to move around and defaecate. This knowledge could be used for future farm husbandry design and promotes the natural behaviour of pigs, thus reducing stress and promoting animal welfare.

**Abstract:**

Little is known on the effect of light on pig behaviour. The choice behaviour of weaned piglets kept under two different light-emitting diode (LED) illuminance levels was investigated: 32 piglets (in two batches) were housed in a preference test room composed of two identical double pen units. One side of the pen unit was permanently illuminated with 600 lux, while the other was darkened to almost 0 lux (~0 lx); by using a passageway, piglets could move between the two sides. The “lying”, “eating” and “activity” behaviours were evaluated during three days in the first, third and fifth experimental week based on video recordings and a 5-min time sampling method. At first, piglets preferred to stay in the 600 lux illuminated compartments. Then, this preference decreased for the “eating” and “activity” behaviours and reversed for the “lying” behaviour, with the darkened compartments being preferred. The results also show that pen soiling was higher under 600 lux, but feed consumption was not affected by the illuminance. Since pigs choose between the two illuminance levels to perform specific behaviours, illuminance could be used to divide the pens into functional areas and, thus, help in meeting pigs’ behavioural needs.

## 1. Introduction

Previous studies have investigated the various direct and indirect effects of light on the animal organism. The results showed that light can influence, among other things, the productivity, e.g., weight gain, feed intake and feed conversion, and general behaviour displayed by pigs [1,2,3,4]. However, care must be taken to ensure that the type of lighting does not have any harmful effects on the animal. Thus, among many other factors, light is suspected to have an influence on the occurrence of biting events [5]. Due to hunting and human disturbance, wild boars had to adjust their activity patterns and are now active during dawn and at night [6]. However, the eyes of these originally diurnal animals have not adapted to low illuminance levels [7].

The illuminance describes the area-related luminous flux that hits an illuminated object. One lux corresponds to a luminous flux of one lumen per square metre (1 lm/m^2^). The SI unit of illuminance is the lux (lx) [8]. Under the open sky on a sunny day, the illuminance can reach values far above 100,000 lx, while in the shade, 10,000 lx only are reached. An overcast winter day reaches around 3500 lx and a full moon night reaches 0.05–0.36 lx [9]. These low values also enable spatial orientation. Information on brightness or minimum illuminance in barns is often given in lux or lumen and varies greatly depending on the publisher of the recommendation or the legal requirements of the respective country. The recommendation for illuminance inside barns ranges from 10 to over 100 lx. In practice, several thousand lux can occur in the area of window surfaces, depending on the incidence of light [10,11,12,13].

The structure and function of the visual apparatus of pigs are similar to that of humans [14,15]; however, different data on pig visual acuity have been reported in the literature. Tanaka et al. [16] described the visual acuity score of pigs as equivalent to that of ruminants, which is 0.045–0.083. In the study of Zonderland et al. [15], the visual acuity of the pigs tested was found to range from 0.017 to 0.07, with smaller visual acuity scores meaning poorer visual acuity. For comparison, the standard visual acuity in humans is 1 [17]. Graf et al. [7] and Zonderland et al. [15] observed that the senses of smell and touch play an overriding role in pig perception and that their vision decreases with decreasing light intensity. Baldwin and Meese’s [18] study on lighting choice behaviour showed that pigs preferred dimmed lighting (10 lx) to bright lighting (110 lx). This is in line with Andersen et al. [19], who found that young pigs preferred darkness, and Taylor et al. [20], who observed that pigs tend to prefer darker compartments (2.4 lx) for resting. However, the results from Tanida et al. [21] are in contradiction, as they found that darkness frightens piglets and that they tend to move towards brightly lit areas, while Larsen [22] observed that the addition of a light (130 lx) above the creep area during darkness did not attract suckling piglets. However, studies in miniature pigs, poultry and rats have shown that constant high intensities of illumination can lead to retinal damage [23,24,25]. Nevertheless, no maximum illuminance levels are listed within the barns, as evidence-based scientific knowledge is lacking [26].

In view of the lack of specificity in lighting recommendations for pigs, the “Innovative light-emitting diode (LED) illumination for increased requirements in livestock animals” project was developed to measure pig behaviours under different LED lighting regimens. Thus, in a previous study [27], the preference choice behaviour of piglets between two colour temperatures (3000 vs. 6500 kelvin) at the same illuminance level (80 lx during the day and 4 lx at night) was investigated. We found that the piglets initially preferred the colour temperature of 3000 K to lie down, but this preference then decreased with time. Additionally, the pigs soiled the pen compartments illuminated with 6500 K to a greater extent, but the LED light colour temperature had no effect on their feed consumption.

In the present study, the aim was to investigate another aspect of LED lighting requirements: the illuminance. Our hypothesis was that piglets choose differently illuminated pen compartments for specific behaviours, which would thus make it possible to divide pens into functional areas according to their behavioural needs. The lying, feeding and activity preference behaviour of weaned piglets for LED-illuminated at 600 lx (and 6500 K) vs. almost 0 lx (~0 lx; i.e., dark) pen compartments was assessed by recording the percentage of piglets under each illuminance pen compartment. In addition, the pigs’ feed consumption and soiling of the pen compartments were measured.

## 2. Materials and Methods

### 2.1. Animals

A total of 32 (16 females and 16 castrated males) 4-week-old weaned piglets of the Large-White × Landrace × Piétrain cross breed were housed in two successive batches in the preference test room of the Martin Luther University of Halle-Wittenberg, Germany. They were brought to the facility directly after weaning from a conventional farm and had only experienced 80 lx light illuminance in pens with windows. In each batch, 8 non-littermate pigs, homogeneously distributed according to their weight, were housed per pen unit (i.e., four females and four castrated males). The piglets weighed, on average, 7.9 ± 1.1 kg at their arrival. Animals had ad libitum access to pellet feed (Mini Start, Denkapig; FA I, Agravis, Querfurt, Germany) from automatic feeders and water from nipple drinkers. The Saxony-Anhalt Regional Administrative Office were notified regarding the experiment but no additional permission was required with regard to the Animal Protection Law (section 7, paragraph 2) because we had a general authorisation from the German Veterinary Office (Nr. TS 17/2018) to keep pigs for experimental purposes in the preference test room of the Martin-Luther University of Halle-Wittenberg as long as no measures inflicting pain, suffering or injury to these animals were carried out.

### 2.2. Experimental Design

For this experiment, the same preference test room as the one described in Götz et al. [27] was used. Briefly, the room comprised 4 identically equipped 4.17 m^2^ pen compartments connected two by two (referred to as pen units) by a passageway (Figure 1). The passageways were 47 cm × 35 cm large and obscured by three hanging opaque black PVC strips (see [27] for more details) in order to separate the two compartments between which pigs could choose. One side of the room was illuminated permanently (24 h) with 600 lx and a colour temperature of 6500 kelvin, while the other half was permanently dimmed to almost 0 lx. Each pen compartment consisted of a concrete slatted floor covering half of the surface, a heating plate (400 mm × 600 mm; MIK International THERMO E, MIK International, Ransbach-Baumbach; Germany) and plastic floor grids covering the other half (Figure 1, [27]). A sisal rope attached to the wall as well as a chain with a piece of soft wood hanging from the ceiling at the height of the piglets were also available in each pen compartment as enrichment for the pigs.

In order to have the two extreme illuminance levels of 600 and ~0 lx, in line with Götz et al. [27], the preference test room was separated in the middle by a wooden framework consisting of squared timber pieces (4 cm high × 2 cm wide). The wooden framework was then covered with opaque fabric (Sunblock blackout fabric; Tuchler; Vienna, Austria). Gaps between the walls and the wooden framework were additionally closed with black tape to prevent light penetration. Likewise, the gaps between the ceiling and wooden framework were also sealed opaque with adhesive tape and strips of fabric. The dark compartments could be reached through an external lockable passageway on the left and right of the test compartments, which allowed animal control in all compartments at any time. Following Götz et al. [27], the required illuminance and the lights’ positions were simulated beforehand to achieve uniform illumination of the 600 lx illuminated compartments. Then, custom-made LED lighting (Schuch 161–162, Adolf Schuch GmbH, Worms, Germany) was installed on a wooden beam at a height of 1.35 m above the first and fourth pen compartments (Figure 1).

The lighting illuminance in the illuminated pen compartments was checked using a Mavolux 5032B luxmeter (Gossen Photo and light measurement GmbH, Nuremberg, Germany). For this purpose, a grid (0.6 m × 0.5 m) of measuring points was laid out on the floor of the compartments (Figure 2). Measurements were taken 20 cm above the floor, which corresponds approximately to the height of the growing pigs’ eyes. Individual measurements were then transferred to an Excel data sheet (Microsoft Corporation, Redmont, Washington, USA) and checked for minimum and maximum illuminance and uniformity of illumination.

The installation height of the lightings was optimised according to these measurements. This measurement process was repeated in the darkened pen compartments. Gaps and cracks within the partition were located and sealed with black tape. In the “bright” pen compartments, an average illuminance of 608.95 lx was achieved with a uniformity of g_1_ = 0.72 (i.e., the value of the ratio between the minimum to average light intensity). This value was slightly higher than the target value of g_1_ = 0.6. This value of illuminance was minimally above the target of 600 lx, but it was within the error tolerance range of the measuring, which was given with a +/− 3% device and was, therefore, accepted. In the “dark” pen compartments, an average illuminance of 0.06 lx with a uniformity of g_1_ = 0.64 could be achieved. The lower uniformity in the dark area was due to the smaller illuminance values. The black polyvinyl chloride (PVC) strips attached to the passageway between the pen compartments reduced the light falling into the dark compartments, but could not completely prevent it. As a result, the measured points in the transition area between the pen compartments showed slightly higher illuminance values than the measured points in the rear part of the compartments.

Overall, a much higher uniformity was achieved in both illuminance levels than the g_1_ = 0.6 required by German national and European standards DIN EN 12464-1 [28], which was used as a guideline value for the test. In order to reduce the effect of the pen compartment position on the pigs’ behavioural preference, the experimental set-up and lightings’ position within the pen units were inversed between the first and the second batch. The two front compartments, which were initially illuminated with 600 lx in the first batch, were darkened (~0 lx) in the second batch.

On the first day of the experiment, the 8 piglets of a pen unit were randomly introduced into either the LED-illuminated (600 lx) or the darkened (~0 lx) pen compartment. Data collection began with the introduction of the last piglet and a final inspection of the pen compartments. Thus, there was no acclimatisation to the pen compartments beforehand. The routine barn work (i.e., animal inspection and cleaning of the pen compartments) took place between 8:00 and 10:00 a.m. daily (see [27] for more details).

### 2.3. Measurements

To investigate the influence of the illuminance level on the preference behaviour of weaned piglets, the pigs’ specific behaviour (i.e., “lying”, “eating” and “activity”), location in the pen (i.e., “light” or “dark” compartment), as well as the feed consumed per pen compartment and the pen cleanliness were recorded. No behavioural observations were made between 8:00 and 10:00 a.m. (i.e., the routine barn work) in order to not bias the results.

Following [27], a video camera (Monacor HDCAM 630, Monacor International GmbH & Co. KG, Bremen, Germany; a 2-megapixel HD-SDI colour camera with day/night function and 2.8–12 mm varifocal lens) was mounted above each pen compartment and recorded the behaviour of the pigs continuously throughout the experiment. Video recordings were stored on a digital recorder (EPHD 08 Everfocus, New Taipei, Taiwan) and were transferred to digital storage media for later analysis. The video recordings of three days (Tuesdays, Thursdays and Saturdays) during the first, third and fifth experimental week were analysed using a 5-min time sampling method [27] in order to define the pigs’ behaviour and location in the pen. The video was paused every 5 min throughout the recorded 24h periods and each behaviour (i.e., “lying”, “eating” and “activity”) and the location (i.e., “light” or “dark” compartment) of each pig were recorded according to a pre-defined ethogram (such as in [27], in a binary way, i.e., 1 = occurring, 0 = not occurring). Following [27], the pig was considered as “lying” when lying down in either a sternal or a lateral position (but no distinction was made for whether the animal was resting or sleeping), was considered as “eating” when it was eating, chewing or had its head above the feed trough (as no distinction could be made), and was considered as engaged in “activity” when neither lying nor eating. 

Based on the method described by the Bavarian State Office for Agriculture for measuring partially slatted floor pens’ cleanliness for fattening pigs [29], pen compartment cleanliness was assessed daily [27]. For this purpose, each pen compartment was artificially divided into four areas (named after their presumed main function: defaecating (area that was kept wet to facilitate the cleaning work), drinking, eating and lying areas) of equal size (Figure 3). The cleanliness of each artificial area of the pen compartments was scored from 0 to 4 as follows: 0 = not soiled; 1 = very lightly soiled; 2 = lightly soiled; 3 = moderately soiled; and 4 = highly soiled (for details, see [27]).

The feed supplied daily was weighed for each pen compartment in order to determine the feed consumption of the pigs. Once a week (on Thursdays), the feed troughs were emptied and the remaining amount of feed in each trough was weighed and subtracted from the weekly feed supplied.

### 2.4. Statistical Analyses

Data were prepared using Microsoft Excel (Microsoft Corporation, Redmont, Washington, USA) and statistical analyses were carried out using Statistical Analysis System 9.4 (SAS Institute Inc., Cary, North Carolina, USA) following Götz et al. [27]. Time sampling behavioural observations of one day were summed beforehand in 2-h sections throughout the day in order to facilitate calculations and reduce the amount of data as in [27].

To evaluate the pig preference behaviour (i.e., feed consumption, lying behaviour, eating behaviour and activity behaviour) under the two different illuminance levels of 600 lx “light” and ~0 lx “dark”, linear mixed models were used, with multiple comparisons being performed using the least-squares means (LSMEANS) statement and the Tukey–Kramer adjustment. Statistical assumptions were checked using a graphical analysis of residuals focusing on the distribution and homoscedasticity of errors of the models.

To examine the feed consumption, a mixed effect model (MIXED procedure) was used. The homogeneity of variances of the data was beforehand verified with a general linear model (GLM procedure) using Levene’s test. For the residual effects, heterogeneous residual variances were modelled (grouped according to the experimental week). Variance components were estimated using the restricted maximum likelihood (REML) method. The “feed consumption” model included the batch (factor with two levels: 1–2), experimental week (factor with five levels: 1–5), pen unit (factor with two levels: 1–2) and compartment illuminance (factor with two levels: 600 lx, ~0 lx). F-tests of overall significance (*p* < 0.05) were calculated in order to retain meaningful variables and interactions only, apart from the compartment illuminance, which was kept. The final “feed consumption” model included the experimental week and the compartment illuminance as fixed effects.

To calculate the probability for the piglets to be under one or the other illuminance, a generalised linear mixed model (GLIMMIX procedure) was used. Proportions under this binomial distribution were logit transformed for the evaluation. Variance components were estimated using the maximum likelihood estimation method. The model included the batch, experimental week, day (factor with six levels: 1–6), time of the day (factor with 11 levels: 0:00, 2:00, 4:00, 6:00, 10:00, 12:00, 14:00, 16:00, 18:00, 20:00 and 22:00), pen unit and their interactions as fixed effects. F-tests of overall significance (*p* < 0.05) were calculated in order to retain meaningful variables and interactions only, apart from the time of the day, which was kept. The final model was calculated for each experimental week (i.e., weeks 1, 3 and 5) and included the day and time of the day only.

To examine the “lying”, “eating” and “activity” behaviours of the piglets, the MIXED procedure was used. Variance components were estimated using the REML method. For the residual effects, heterogeneous residual variances were modelled (grouped according to the pen unit). The models included the same fixed effects as the illuminance preference model. F-tests of overall significance (*p* < 0.05) were calculated in order to retain meaningful variables and interactions only, apart from the compartment illuminance and time of the day, which were kept. The final models were calculated for each experimental week (i.e., weeks 1, 3 and 5) and included the day, time of the day, compartment illuminance and the time of the day × compartment illuminance interaction.

A Chi-square test was used to compare the pen compartment cleanliness score distribution under the two illuminance levels (i.e., 600 lx and ~0 lx). Because the “drinking”, “eating” and “lying” areas were not or only slightly soiled, the scores of the “defaecating” area only were considered to assess the overall pen compartment cleanliness. This prevented an evaluation using the Chi-square test with too many discarded, empty classes.

## 3. Results

### 3.1. Illuminance Preference

By using the time-sampling method, 76,032 behavioural observations were recorded in total, including 31,703 observations (about 41.7% of all observations) made under 600 lx and 44,329 (about 58.3% of all observations) made under ~0 lx (Table 1). With 88.37% of all observations, the “lying” behaviour was the most frequently observed in both batches and under both illuminance levels. The “eating” and “activity” behaviours were shown in 5.24% and 11.39% of all observations, respectively. 

In the first experimental week, the pigs showed a preference for spending more time under 600 lx throughout the day (*p* ≤ 0.006; Figure 4a), with, on average, 77.6% of the pigs staying under this illuminance.

In the third experimental week, the animals’ preference for illuminance changed (*p* ≤ 0.001; Figure 4b), with 77.5% of the pigs staying under ~0 lx. 

In the fifth experimental week, the preference of the pigs for the darkened compartments decreased (*p* ≤ 0.03; Figure 4c), but was still clear, except at 16:00, where no illuminance preference was detected (*p* = 0.79). 

Throughout the experimental weeks, an effect of the day on the illuminance preference was observed (*p* < 0.001; Table 2). 

### 3.2. “Lying” Behaviour

With on average 83.70% of the pigs lying down (Figure 5), the “lying” behaviour was the most performed behaviour. The illuminance (*p* < 0.001), time of the day (*p* < 0.001) and their interaction (*p* < 0.001) had an effect on the “lying” behaviour throughout the experiment. The day of observation also had an effect during the first experimental week (*p* < 0.001), but not during the third (*p* = 0.07) and fifth experimental weeks (*p* = 0.21).

In the first experimental week, pigs preferred to lie down in the compartments illuminated with 600 lx (Figure 5a), except between 4:00 and 6:00, where the pigs showed no preference (*p* = 0.55). The illuminance preference of the pigs reversed from the third experimental week to the darkened compartments, which were preferred at all times (*p* < 0.001; Figure 5b). The pigs maintained this preference for the darkened compartments in the fifth experimental week (Figure 5c), except between 18:00 and 20:00, where the pigs showed no preference (*p* = 0.54). 

### 3.3. “Eating” Behaviour

The “eating” behaviour was shown, on average, by 5.24% of the pigs at the time.

In the first experimental week (Figure 6a), the time of the day (*p* < 0.001), day (*p* < 0.001), illuminance (*p* < 0.001) and time of the day × illuminance interaction (*p* < 0.001) had an effect on the “eating” behaviour. Between 4:00 and 18:00, the pigs preferred to eat in the brightly illuminated compartments. 

In the third experimental week (Figure 6b), the time of day (*p* < 0.001) and illuminance (*p* < 0.001) had an effect on the “eating” behaviour. However, the day (*p* = 0.08) and the time of the day × illuminance interaction (*p* = 0.84) had no effect. A preference for the darkened compartments could be observed at 2:00, 18:00 and 22:00.

In the fifth experimental week (Figure 6c), the time of the day (*p* < 0.001) and the time of the day × illuminance interaction (*p* < 0.001) had an effect on the “eating” behaviour. However, the day (*p* = 0.55) and the illuminance (*p* = 0.84) had no effect. Overall, no clear preference for one or the other illuminance could be observed throughout the day. However, at 10:00, the pigs preferred the 600 lx lit compartments, while at 20:00, they preferred to eat in the darkened compartments.

### 3.4. “Activity” Behaviour

On average, 11.39% of the pigs were active at the same time, mainly between 6:00 and 20:00.

In the first experimental week (Figure 7a), the time of the day (*p* < 0.001), day (*p* = 0.001), illuminance (*p* < 0.001) and time of the day × illuminance interaction (*p* < 0.001) had an effect on the “activity” behaviour. From 4:00 to 8:00, and at 12:00 and 16:00, the pigs showed a preference to be more active in the brightly lit compartments.

In the third experimental week (Figure 7b), the time of the day (*p* < 0.001), day (*p* < 0.001), illuminance (*p* < 0.001) and time of the day × illuminance interaction (*p* = 0.012) had an effect on the “activity” behaviour. The piglets were particularly active between 06:00 and 22:00 and showed a preference to behave more actively in the pen compartments illuminated with 600 lx at 06:00, 14:00 and in the evening at 18:00.

In the fifth experimental week (Figure 7c), the illuminance (*p* = 0.10) had no effect on the “activity” behaviour. On the contrary, the time of the day (*p* < 0.001), day (*p* < 0.001) and time of the day × illuminance interaction (*p* = 0.012) had an effect on the “activity” behaviour. At 06:00, the animals showed a preference to show activity in the darkened pen compartments, while at 16:00, they preferred being active under the 600 lx illuminance. In the other observation periods, no further clear preference could be found.

### 3.5. Pen Compartment Cleanliness 

Throughout the experiment, the compartment illuminance was found to have an effect (x^2^_3_ = 135.94, *p* < 0.001; Figure 8) on the pen compartment cleanliness scores. Pen compartments illuminated with 600 lx were more often polluted (scores 2 and 3) than the darkened pen compartments (~0 lx), where less or no soiling (scores 0 and 1) was measured.

### 3.6. Feed Consumption

The pigs consumed 6.7 ± 1.5 kg (LSM ± SE) of rearing feed in the first week, 14.2 ± 1.8 kg of sole food in the second week, 21.5 ± 4.7 kg in the third week, 25.3 ± 4.9 kg in the fourth week and 35.5 ± 9.1 kg in the fifth week.

The illuminance (*p* = 0.86) had no influence on the amount of feed consumed, with 20.25 ± 3.3 kg of feed eaten under 600 lx and 21 ± 3.3 kg under ~0 lx. The light intensity × experimental week interaction (*p* = 0.228) also had no effect on feed consumption, whereas the experimental week alone (*p* = 0.004) had an effect on the amount of feed consumed by the pigs.

## 4. Discussion

The aim of the study was to investigate whether piglets have a preference for the 600 lx (and 6500 kelvin) or almost 0 lx illuminance to perform specific behaviours. Over 5 weeks, the percentage of pigs lying, eating and being active under each illuminance was recorded and evaluated. The amount of feed consumed as well as the pen compartment soiling were also measured for each illuminance. In the first experimental week, the pigs showed a preference for the pen compartments illuminated with 600 lx, where all measured behaviours were more frequently performed. In the third and fifth experimental weeks, this preference changed and the pigs spent more time in the darkened pen compartments, especially when lying down. The degree of pen compartment soiling was also influenced by the illuminance, with pen compartments lit with 600 lx being more soiled than the darkened pen compartments. The amount of feed consumed was not influenced by the illuminance.

In the present study, the pigs showed a clear preference for the 600 lux illuminance in the first experimental week, with, on average, more than 2/3 of the pigs staying in the brightly lit pen compartments. This is in agreement with the study of Tanida et al. [21], which showed that 1-week-old suckling piglets actively move towards bright areas and actively move away from dark areas. On the contrary, Parfet et al. [30] observed that newborn piglets tend to be attracted to darker areas. However, it should be pointed out that, in these two studies, the age (much younger piglets) and experience of the animals with their environment were different from the present study. In their motivational study with 14-week old pigs, Baldwin and Meese [18] found that the pigs strongly preferred light over darkness, spending at least 72% of the time in light. As already seen in rats, it is likely that the pigs initially preferred the compartment that corresponded to the familiar environment in which they were born and raised and, therefore, opted for the brightly lit pen compartments [31]. In the third experimental week, however, this lighting preference changed, with more than 3/4 of the pigs staying in the darkened pen compartments throughout the day. Towards midday, the proportion of pigs in the brightly lit areas increased, but decreased again towards evening. These observations are similar to those of Hacker et al. [32] who found that growing pigs that could constantly choose between a lit and a dark room spent most of the day (75%) in the dark. In the fifth experimental week, the proportion of pigs in the brightly lit pen compartments increased slightly, with no more illuminance preference detected at 18:00. Still, almost 2/3 of the pigs preferred to stay in the darkened pen compartments. A reasonable explanation for this could be the relative decrease in space availability (see also [27]) due to the increasing size of the growing pigs (2.0 kg of live weight/m² in the first vs. 5.30 kg of live weight/m² in the fifth experimental week). The same phenomenon was also observed in our previous study investigating pigs’ colour temperature preferences [27]. Some of the pigs stayed in less preferred areas in the fifth experimental week, as their growth increased and the space available was limited.

The “lying” behaviour was the most performed behaviour, with pigs lying down or resting about 83% of the time. Accordingly, Sambraus [33] and Marx [34] observed that conventionally kept pigs spent between 80% and 90% of the day lying down. Similar results were also found in our colour temperature preference study [27], with almost 86% of the pigs lying down. Thus, concurring with the global illuminance preference, pigs showed a preference for lying down in the brightly lit (600 lx) pen compartments during the first experimental week only and in the darkened pen compartments from the third experimental week onwards. Taylor et al. [20] also observed that weaned piglets spent most of their resting time in the darkest compartment (i.e., 2.4 lx illuminance vs. 4 lx vs. 40 lx vs. 400 lx) when given the choice. Therefore, pigs should be provided with a sufficient darkened space to rest.

The “eating” behaviour represented the lowest percentage of all the behaviours recorded. This differs, however, from wild boars who spend more time eating and exploring [35]. In the first experimental week, the pigs were eating more often in the brightly lit compartments than in the darkened pen compartments. This may have resulted from the fact that the pigs generally spent more time in the brightly lit compartments and, thus, ate there too. In the third experimental week, this preference could only be observed at certain hours of the days, until no clear illuminance preference could be observed in the fifth experimental week. This is in agreement with the study of Taylor et al. [20] in which pigs showed no preference to eat in 2.4, 4, 40 or 400 lx illuminated areas. For this behaviour, pigs clearly differ from other animal species such as cows, sheep or goats, which prefer not to eat in darkness or to eat much less food [34,35,36]. Further evidence that the illuminance had no effect on pigs’ “eating” behaviour is the fact that approximately the same amount of feed was consumed under both illuminance levels. As regards the feed consumption, the only notable and logical effect observed was the increased amount of feed consumed over the experimental weeks, i.e., in correlation with the growth of the pigs.

Overall, pigs showed phases of increased activity, especially in the morning hours (from 6:00) and in the afternoon, though this activity was found to represent approximately only 1/10 of all the behaviours recorded. As reported by Tilger [35], the main activity phases of wild boars are dawn and dusk; however, in domestic pigs, management factors and, in particular, the feeding time or—in our case—the animal inspection and pen cleaning, have a strong influence on the activity periods. Zaludik [36] also found that over an observation period of 7 h, pigs on partially or fully slatted floors showed active behaviours only up to 8.8% of the time and spent the rest of the time lying down. As regards the illuminance-based preference to perform the “activity” behaviour, no clear preference could be found throughout the experiment, even though it occurred more often in the brightly lit pen compartments, especially in the first and third experimental weeks. Although it was not measured separately, this active behaviour was often accompanied by play behaviour of several pigs that chased each other through the whole pen. This led to observations in otherwise less frequented pen compartments as piglets used all the space they had available regardless of the illuminance level. In addition, times of the day when the “activity” behaviour was recorded often corresponded to times of the day when pigs were observed eating. Sometimes, pigs were observed moving from one feeding trough to the feeding trough in the other compartment, i.e., with another illuminance level, if they could not obtain the opportunity to eat where they first tried. Displacements when trying to eat at a trough were also classified as active behaviour. It can be surmised that a different feeding system or a smaller animal:feeding-place ratio (here of 4–5 piglets:1 at the beginning and 2–3 piglets:1 later on) would have resulted in decreased activity during the feeding periods and more pigs could have been clearly observed eating under a specific illuminance.

The degree of soiling of the pen compartments was found to depend on the illuminance level. Pen compartments illuminated with 600 lx were more often and heavily soiled than the darkened pen compartments. In the third and fifth experimental weeks, pigs rested more in the darkened compartments and used the other compartments, the brightly lit ones, to defaecate. Thus, pigs seem to avoid eliminating where they lie if the conditions allow it [37,38]. These findings are in accordance with the studies of Opderbeck et al. [39] and Taylor et al. [20] who found that pigs preferred to defaecate in the brightly lit areas. The ability to minimize soiling to specific areas of the pen offers advantages in terms of air quality in the area where the pigs are staying, as well as manure management. Furthermore, distinct resting and manure areas, in addition to allowing the expression of natural behaviour for the pigs, is beneficial for their health. Thus, it seems that light intensity could help to create targeted defaecation areas [40,41].

As in our previous experiment [27], an overall effect of the day of the observations was observed in the first experimental week on the pigs’ illuminance preference for all the recorded behaviours and throughout the experiment for the pigs’ preference location in the pen and their “activity” behaviour. Throughout the experiment, care was taken to keep the conditions in the pen compartments constant. This included changing the cleaning order of the pen compartments daily and always trying to mark the pigs in different locations within the pen. Nevertheless, the pigs did not behave the same from day to day. By selecting three different observation days per experimental week as in [27], we wanted to obtain a representative overview of the pigs’ behaviour. Evaluating more days per week or using a different evaluation method could possibly reduce the influence of the day, but would be accompanied by an increased evaluation effort.

Finally, it remains uncertain whether 600 lx is an ideal illuminance for pigs, as opposed to almost 0 lx. This illuminance level (600 lx) was chosen because it corresponds to the value that often occurs in practice near windows. Furthermore, a higher illuminance level may lead to more aggression in pigs (as seen in an unpublished study). Follow-up experiments should, therefore, investigate other illuminance levels but also compare the behaviour of pigs kept under artificial vs. natural light.

## 5. Conclusions

The current study shows that pigs have illuminance preferences to perform certain behaviours (i.e., lying, eating, being active and eliminating) and that these preferences varied across the experimental weeks. In particular, if pigs initially preferred to lie down in the brighter pen compartments (lying being the most performed behaviour), this preference quickly reversed for the darkened compartments. On the other hand, pigs eliminated and were active predominantly in the 600 lx illuminated pen compartments. No effect of the illuminance was found on the feed consumption. To find out whether the behaviour of pigs can be easily guided into specific functional areas by using different illuminance levels and/or colour temperatures, further investigations should be carried out. This would allow clearer delimitations of functional areas in the pen that correspond to the pigs’ natural behaviour and would benefit their well-being and health. Moreover, attention should be paid to the housing density, so that pigs always have the choice to perform their natural behaviour as, when and where they wish. However, this is only one way of enriching pigs’ environments and does not replace the discussion on whether animal welfare can be improved in a more sustainable way by increasing the space available or by providing outdoor access.

## Figures and Tables

**Figure 1 animals-12-00202-f001:**
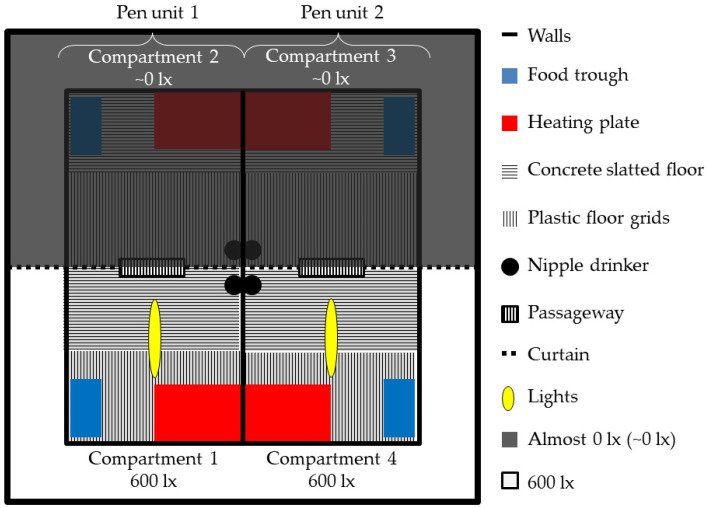
Design of the preference test room and pen compartments.

**Figure 2 animals-12-00202-f002:**
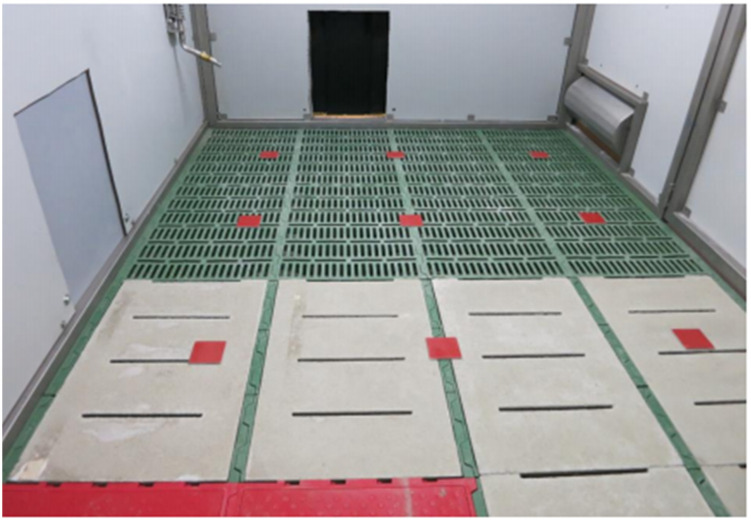
Pen compartment with measuring grid laid out (red metal plates).

**Figure 3 animals-12-00202-f003:**
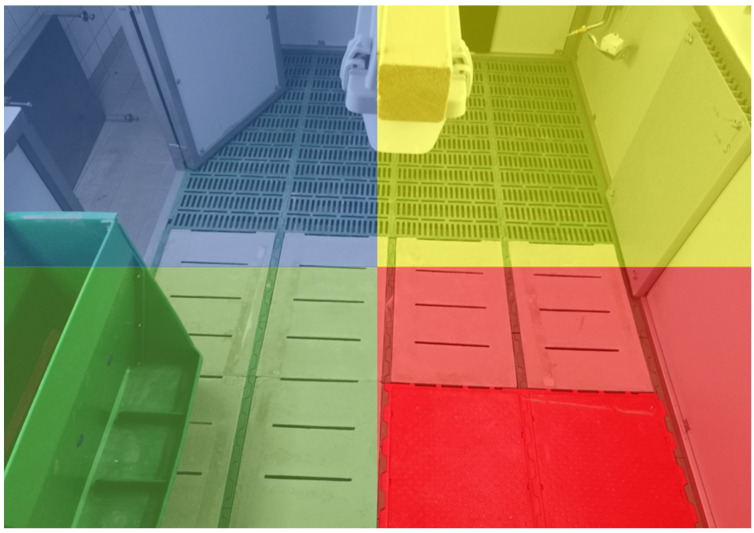
Artificial partitioning of the pen compartment into four functional areas: defaecating (blue), drinking (yellow), eating (green) and lying (red) areas.

**Figure 4 animals-12-00202-f004:**
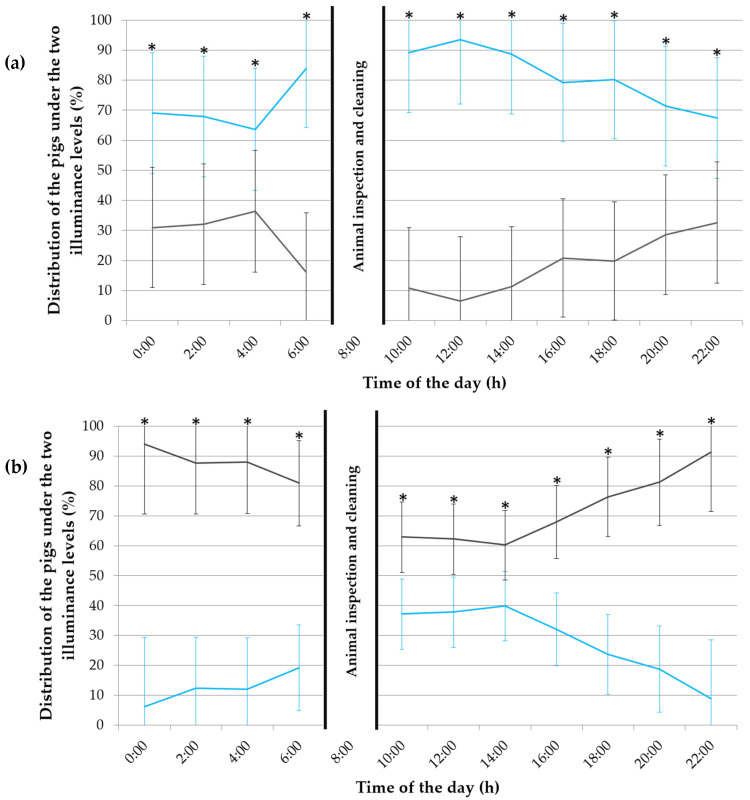
Distribution of the pigs (in %) throughout the day (24 h) under the 600 lx (in blue) and almost 0 lx (in grey) illuminance during (**a**) the first, (**b**) third and (**c**) fifth experimental week. The error bars represent the standard error (±SE) of the mean. An asterisk (*) indicates a statistical difference with *p* < 0.05.

**Figure 5 animals-12-00202-f005:**
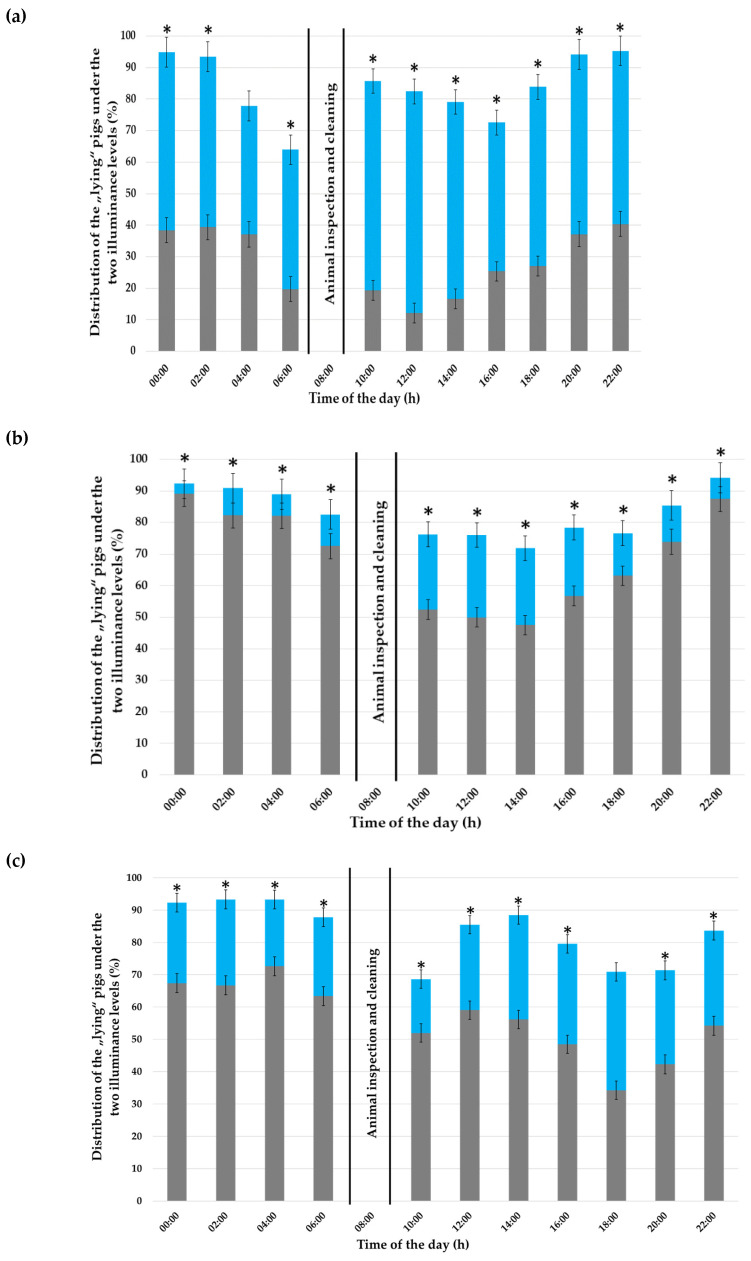
Distribution of the “lying” pigs (in %) throughout the day (24 h) under the 600 lx (in blue) and almost 0 lx (in grey) illuminance, given as percentages of the total performed behaviours, during (**a**) the first, (**b**) third and (**c**) fifth experimental week. The error bars represent the standard error (±SE) of the mean. An asterisk (*) indicates a statistical difference in the time of the day × compartment illuminance interaction with *p* < 0.05.

**Figure 6 animals-12-00202-f006:**
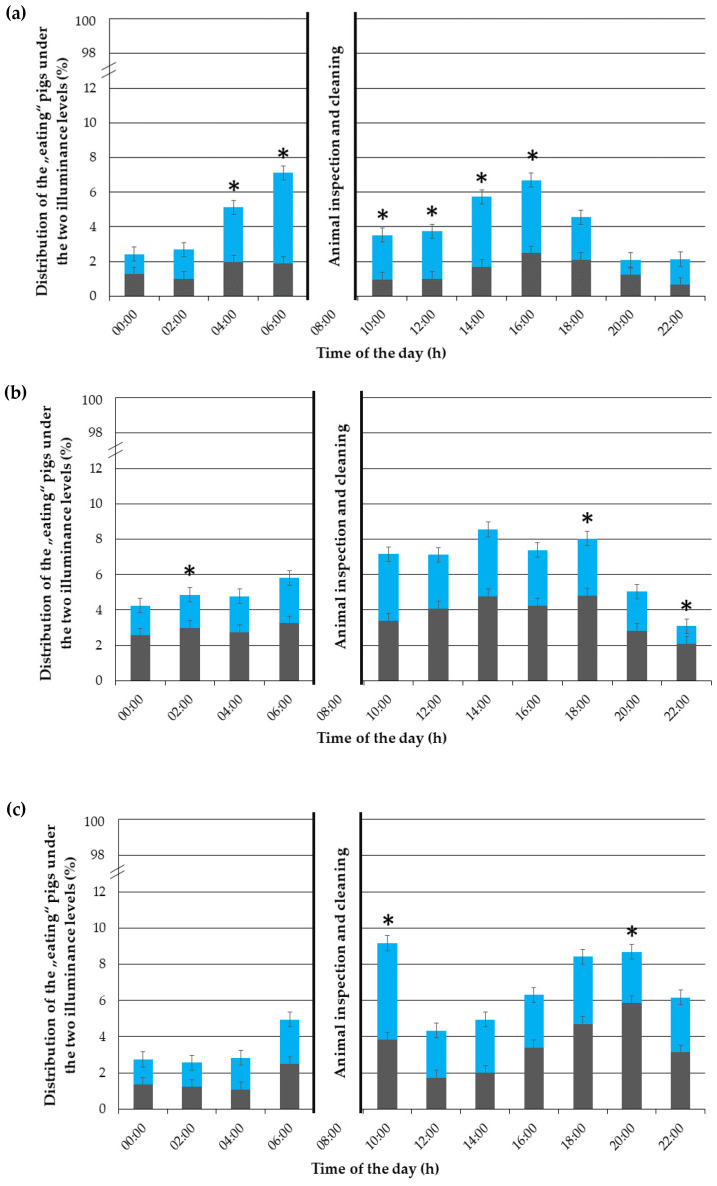
Distribution of the “eating” pigs (in %) throughout the day (24 h) under the 600 lx (in blue) and almost 0 lx (in grey) illuminance, given as percentages of the total performed behaviours, during (**a**) the first, (**b**) third and (**c**) fifth experimental week. The error bars represent the standard error (±SE) of the mean. An asterisk (*) indicates a statistical difference in the time of the day × compartment illuminance interaction with *p* < 0.05.

**Figure 7 animals-12-00202-f007:**
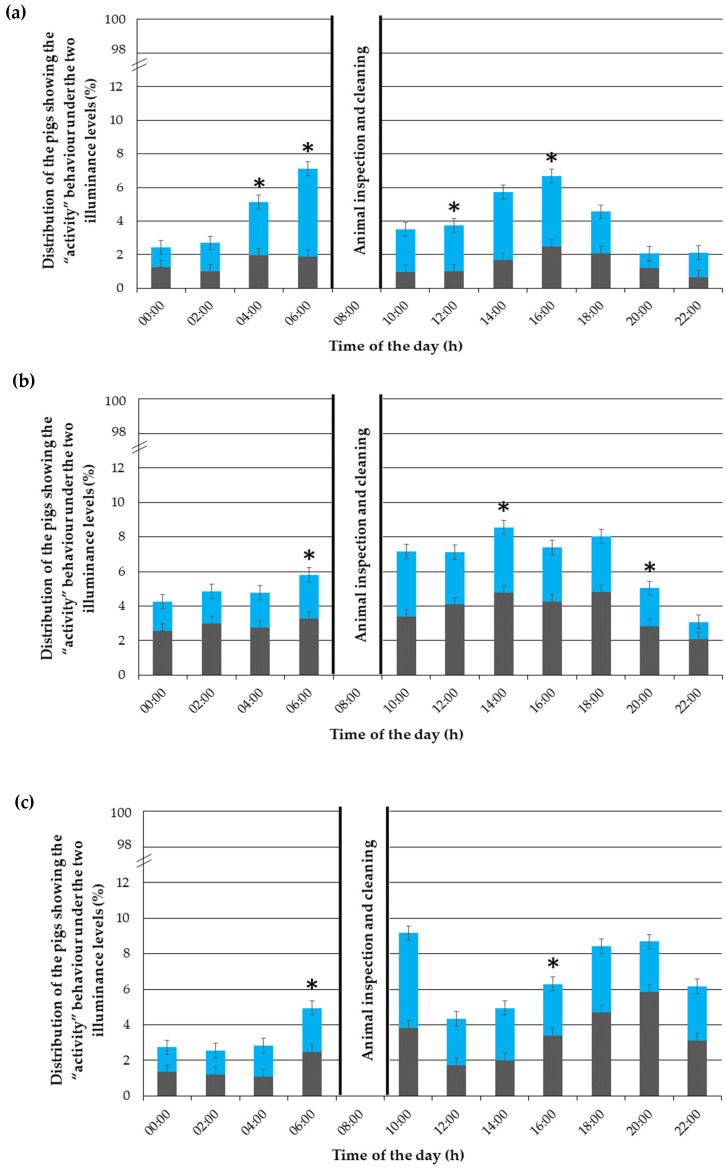
Distribution of the “activity” pigs (in %) throughout the day (24 h) under the LED the LED light illuminance of 600 lx (in blue) and almost 0 lx (in grey), given as percentages of the total performed behaviours, during (**a**) the first, (**b**) third and (**c**) fifth experimental week. The error bars represent the standard error (±SE) of the mean. An asterisk (*) indicates a statistical difference with *p* < 0.05.

**Figure 8 animals-12-00202-f008:**
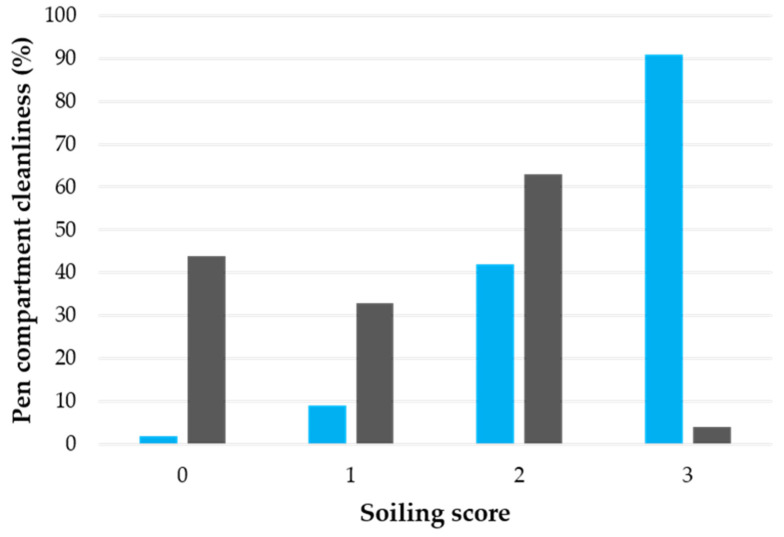
Pen compartment cleanliness score distribution (in %) under the 600 lx (in blue) and almost 0 lx (in grey) illuminance. The following cleanliness scores were used: 0 = not soiled; 1 = very lightly soiled; 2 = lightly soiled; 3= moderately soiled; and 4 highly soiled.

**Table 1 animals-12-00202-t001:** Distribution of the behaviours observed (in %) under the 600 lx and almost 0 lx (~0 lx) illuminance levels.

Illuminance	Observed Behaviours (%)			
	“Lying”	“Eating”	“Activity”	Total
600 lx	32.28	2.68	6.74	41.7
~0 lx	51.09	2.56	4.65	58.3
Total	83.37	5.24	11.39	

**Table 2 animals-12-00202-t002:** Percentage of pigs observed for each recorded day (Days 1–3) of the first, third and fifth experimental week (with batches 1 and 2 summed) under the 600 lx and almost 0 lx (~0 lx) illuminance.

Illuminance	Pigs Observed (%)								
	Week 1			Week 3			Week 5		
	Day 1	Day 2	Day 3	Day 1	Day 2	Day 3	Day 1	Day 2	Day 3
600 lx	64.3	67.7	66.9	25.2	20.5	22.9	30.6	36.6	40.6
~0 lx	35.7	32.3	33.1	74.8	79.5	77.1	69.4	63.4	59.4

## Data Availability

The datasets analysed during the current study are available from the corresponding authors on reasonable request.

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
