# Peer review of "Lying, Feeding and Activity Preference of Weaned Piglets for LED-Illuminated vs. Dark Pen Compartments"

_animals, 2022, doi:10.3390/ani12020202_

Round 1

Reviewer 1 Report

General comments

I thank the authors for a well written and interesting manuscript – lighting is often an overlooked resource although whilst an expensive change to make in an existing animal unit, it is important to have a scientific evidence base for newly designed units. I wonder how achievable a strong light differential might be in an open pig pen, with other adjacent pens similarly lit – an obvious follow up experiment would be to study the effects seen here in a standard environment.

I found the English and writing style easy to follow, and the statistics and methodology robust and well presented. I noted some specific issues that are outlined below for consideration. I don’t believe any of these comments represent a major challenge to publication.

Specific comments

Line 21 – given the authors are using English English spellings (e.g. behaviour) then is the correct spelling “defaecation”, following “faeces” used correctly in other places in the manuscript?

Line 43 – the use of a single quotation mark in large numbers e.g. 100’00 here is consistent but an unusual convention? 100,000 etc?

Line 48 – the word stable (line 186 also) to describe a pen shed/building is technically acceptable but surprising unusual to this native English speaker. I immediately think of horses.

Line 59 – odd formatting to correct, caused by justification to both margins I believe. A return at the end of the sentence should fix it

Line 60-65 – not sure this detail is really needed although it is not detracting from the manuscript if it stays

Line 75 – the use of furthermore is this sentence indicates a piece of evidence in addition to that previously supplied, whereas I think the sentence is indicating a contrast. I wonder if a “however” might flow better?

Line 107 – the issue of animal ethics is an important one, especially given the influence of pig welfare that the authors have described. It would appear from this sentence that local regulations have been adhered too; I know in my locality and institution an animal ethics application would be required.

Line 125 – When reading the discussion around the various chosen locations, I looked more closely at figure 1 and noticed that the drinkers and feeding troughs are by necessity placed on different flooring types in the two halves of the pen. Flooring type can influence pig behaviour (choice of defaecation location for example) as does the wet area created around drinkers and this is a small confounding issue unfortunately.  I’m sure not a major issue or one that significantly affects the work but an unfortunate niggle.

Line 129 – I am having difficulties visualising the description here. The scaffolding was 4cm high? Or is that in fact the dimensions of the timber pieces used?

Line 152 – “lightings”? Either lighting or lights is better?

Line 245 – use of the word “inevitably” (also line 255). Often a word that has a more negative connotation such as “unavoidable”; I assume the authors mean that these terms were kept in the model as key components – I’m not sure what word might be better but probably there is no need to use an adverb at all?

Line 291 – figure 4 a line graph in comparison to the following bar graphs. Could fig 4 also be a bar graph? Consistency? Not a major issue either way really, it just stands out a little. Also the use of “repartition” in the axis titles in the charts – it would indicate a further partitioning whereas really we are looking at the partition of animals across the space?

Line 340 – figure 6. Large y axis seems to be dictated by the axis title rather than the spread of the data. Axis goes up to 30% but data does not up above 10%. Ditto figure 8.

Line 366 – What happened to figure 7? Appears to have got lost between figures 6 and 8?

Author Response

Thank you for your revisions, please find our answers as a word document 

Reviewer 2 Report

I think that this is a worthy study that should be published. I think that the introduction, results and discussion could be improved, so that people understand the relevance of this study, and are able to better put it into a broader context of how pigs behave under different lighting, natural lighting, feral conditions etc. There are some references made to these points, but more could still be done. I also think that there are some minor methodological issues that require clarification (e.g. reason for imbalanced observation hours of the light and dark conditions?). There are also some minor grammatical errors throughout, many of which could be corrected by using a good editing tool, such as the microsoft word editor. I have pointed out a few of these issues, and make more specific comments and suggestions for improvement below:

Line 19- “a free choice between two..” or “piglets could choose between two…” as written now it is redundant.

Abstract- I would add one sentence at the beginning that includes the rationale for this study, as you did in your simple summary.

Lines 28-30, I guess its not specific behaviors that were video recorded, but rather the entire pen/group etc, and then the videos were analyzed to score lying, eating and activity, right? Clarify this. Also please further clarify what you mean by time sampling-was this whole interval recording, partial interval recording or momentary time sampling?

Line 33- “pen’s soiling”? Pens can not soil themselves. Use either “pen soiling” or “soiling of pens”

Introduction- I would begin first with some background on pig welfare and importance of light conditions more generally (maybe move Line 51 material to the top). Get people interested in the topic before you go into the details on illuminance, which are quite interesting, but we need context first.

Lines 51-58, are any of these studies looking at natural lighting (e.g. through a window) or are they all about artificial lighting? Have any studies found effects of natural vs artificial lighting on behavior and welfare? This would be interesting to report.

Line 82: “thus in a previous study….” (remove the word ‘own’)

Line 85: “decreased with time” (remove the word ‘the’)

Line 173- What housing did the piglets have before they were introduced to the experimental pen? Was they introduced directly at weaning, or had they already been weaned, and if so had they been housed in a smaller space or a space of the same size as the experimental pen, and with what type of lighting? All of this may be relevant to how pigs first react and why their preference changes over time. Were they littermates or mixed groups? If mixed, how long had they been mixed? We need more information here.

Lines 207-208- I do not understand how the different areas (defecating, eating, drinking, lying) were demarcated- was this based on how the pigs used them, or rather was each area designed differently? This is not clear and could influence the results.

Lines 276-278. Can you explain why you didn’t take an exactly equal amount of observations under 600 lx and 0 lx? It is surprising that you didn’t use a balanced number of observations, as this would have been possible. It also makes Table 1 harder to interpret. For example, it appears on Table 1 like pigs spend a lot less time laying in the 600 lx vs 0 lx, but actually since the 600 lx was observed much less often, they are spending a similar proportion of total observation time laying in each of the two conditions (e.g. 32.28/41.7 vs. 51.09/58.3). This is precisely why it would have been better to observe them for equal amounts of time, or at least correct for this better in your results.

Table 1.  I would love to see in the discussion some indication that these values (under both conditions) while likely typical of captive pigs (I assume that they are) also are totally different of what wild or feral pigs would do, where I assume a much greater proportion of time would be spent feeding and active. I know it is not the point of your paper, but it is an important point to make, that above and beyond the lighting options, there is a bigger issue here.

Figure 4. Are these variances due to individuals or is it variance between the two groups? Would be good to clarify that.

Table 1 and Table 2, when summing across batches and presenting percentages, we can not get a sense of how much variation there is between the batches. It might be nice to find a way to report this too.

Figure 5. Are these means and SEs for the batches? For individuals? Please clarify.

Line 397- “lying, eating and being active”

Lines 409-411. Seems strange to compare weaned pigs to one week olds. But maybe there is just little` other data on light/dark preferences of older pigs?

Lines 416-419. We need the information about the previous familiar compartment (size of pen, lighting, social groups  etc) to be included in the methods as I mentioned above.

Lines 427-433. If you collected data on dominance relations of the pigs (e.g. based on aggression at weaning), you could test this hypothesis by checking whether subordinate pigs are ‘forced’ to stay in less preferred areas to space out more.

Line-443. Or… therefore we have a long way to go to provide pigs with housing environments that can meet even their most basic needs for movement, exploration and stimulation! I know its not the point of your study, but seems like a good opportunity to talk about pigs needs that are not met by changes in lighting.

Line 470- separately

Conclusions and in general- I am not sure from your set up if the avoidance of the 600 lx compartment is because dark is preferred, or rather because it was somehow too bright (e.g. not ideal light for the pigs). It would be very interesting to compare this with a situation where pigs can get natural light from one side of the cage and darkness in the other. I wonder if the results would be similar? Or if you varied light intensity would you get different results? It would be nice to remind the reader why you picked 600 lx and if you are convinced it is the ideal lighting for pigs, or should other illuminances be tried as well compared to dark to better determine pig preferences?

Author Response

(The authors gave the same response as above.)

Reviewer 3 Report

You dit not mention wether the piglets were offered manipuable material.

Author Response

(The authors gave the same response as above.)
